# Current Applications and Future Perspectives of Artificial and Biomimetic Intelligence in Vascular Surgery and Peripheral Artery Disease

**DOI:** 10.3390/biomimetics9080465

**Published:** 2024-08-01

**Authors:** Eugenio Martelli, Laura Capoccia, Marco Di Francesco, Eduardo Cavallo, Maria Giulia Pezzulla, Giorgio Giudice, Antonio Bauleo, Giuseppe Coppola, Marco Panagrosso

**Affiliations:** 1Division of Vascular Surgery, Department of Surgery, S Maria Goretti Hospital, 81100 Latina, Italy; eugenio.martelli@uniroma1.it; 2Department of General and Specialist Surgery, Sapienza University of Rome, 00161 Rome, Italy; 3Faculty of Medicine, Saint Camillus International University of Health Sciences, 00131 Rome, Italy; 4Division of Vascular and Endovascular Surgery, Department of Cardiovascular Sciences, S. Anna and S. Sebastiano Hospital, 81100 Caserta, Italy

**Keywords:** artificial intelligence, biomimetic intelligence, peripheral arterial disease, vascular surgery, artificial neural network, convolutional neural network

## Abstract

Artificial Intelligence (AI) made its first appearance in 1956, and since then it has progressively introduced itself in healthcare systems and patients’ information and care. AI functions can be grouped under the following headings: Machine Learning (ML), Deep Learning (DL), Artificial Neural Network (ANN), Convolutional Neural Network (CNN), Computer Vision (CV). Biomimetic intelligence (BI) applies the principles of systems of nature to create biological algorithms, such as genetic and neural network, to be used in different scenarios. Chronic limb-threatening ischemia (CLTI) represents the last stage of peripheral artery disease (PAD) and has increased over recent years, together with the rise in prevalence of diabetes and population ageing. Nowadays, AI and BI grant the possibility of developing new diagnostic and treatment solutions in the vascular field, given the possibility of accessing clinical, biological, and imaging data. By assessing the vascular anatomy in every patient, as well as the burden of atherosclerosis, and classifying the level and degree of disease, sizing and planning the best endovascular treatment, defining the perioperative complications risk, integrating experiences and resources between different specialties, identifying latent PAD, thus offering evidence-based solutions and guiding surgeons in the choice of the best surgical technique, AI and BI challenge the role of the physician’s experience in PAD treatment.

## 1. Introduction and Background

Artificial Intelligence (AI) made its first appearance in 1956 and has progressively gained relevance in our lives, not only in media, telecommunications, marketing, transport, finance, education, sport but also in healthcare systems, and patients’ information and care. AI’s role is to simulate human intelligence capacity in quite every process, especially in understanding the most basic aspects of living organisms so as to transfer those properties to human applications. Biomimetic intelligence (BI) applies the principles of systems of nature to create biological algorithms, such as genetic and neural network, to be used in different scenarios. In particular, their properties can be used to exploit biologically inspired designs in technology, biomedicine, and engineering systems, and are mostly useful in healthcare. BI is specifically programmed to develop human capacities that can be transferred to daily activities. 

Programming BI includes focusing on cognitive skills such as:-Learning: acquiring data and creating rules and roads with the aim of transforming them into actionable information. Those roads or rules are called algorithms, and their work is to guide computing devices to accomplish a specific task by step-by-step instructions;-Reasoning: combining the right algorithms to accomplish the desired task, to perform the specific role;-Self-correction: a specific capacity of self-testing that allows for continuous corrections and refining to obtain the best outcome in time;-Creativity: the most stunning and human-emulating BI capacity. It is based on neural networks, rules-based systems, statistical methods,+--+-+ and other BI techniques used to create images, texts, music, etc.

AI (and BI in particular) functions can be grouped under the following headings: Machine Learning (ML) is able to predict output given new input data by using a machine-induced process of learning; Deep Learning (DL) is a specific class of ML that uses multiple processing layers to amplify input data and suppress irrelevant variations to increase the probability of detection and classification; Artificial Neural Network (ANN) and Convolutional Neural Network (CNN) are the most commonly used DL algorithms, both inspired by human neural connections organized in layers; Computer Vision (CV) is the highest representation of imaging detection and elaboration for daily problem-solving [1]. 

They aim to replicate human capacities of understanding, learning, remembering, reasoning, adapting, interacting, and, finally, creating new solutions, basically incorporating and combining a big amount of data in a mathematical relationship to create a model that can predict new values applied to new data. The unvaluable advantage of AI and BI is the capability of processing large amounts of data much faster than human researchers and making predictions more accurately than possible in human processes. Machine learning artificial intelligence applications can provide quick and useful information starting from huge databases whose analysis would be extremely time-consuming and prone to errors. On the other hand, they lack the ability to generalize from one task to another that remains, nowadays, a human intelligence prerogative. Data analysis in thehealthcare system is of the utmost importance so that the AI and BI roles in this field are well established and acknowledged today. The most important role of AI and BI in healthcare is to improve patients’ outcomes. Faster and more accurate diagnoses are often performed by AI employment in those days. An example of that is artificial technologies used to predict, understand, and contrast the COVID-19 pandemic. Another example is the use of artificial and biomimetic intelligence to review and analyze mammograms in a faster and more accurate way in order to reduce the need for unnecessary biopsies. Drug research and discovery is nowadays an important area of artificial and biomimetic intelligence application in healthcare. Thanks to the possibility of swiftly directing the most recent advances in drug discovery to new drugs’ production and circulation, health science progress could improve patients’ treatment. 

In this way, BI can contribute to reducing healthcare system costs. Specific BI technologies are capable of understanding natural language and answer its questions by combining patients’ and environment data to form a hypothesis accompanied by a confidence scoring set. The sensitive increase in consumer wearables and other medical devices allows for a good and reliable way of database filling to work on for BI algorithm construction, as well as for monitoring or detecting potential life-threatening heart or cerebral disease episodes. Online or virtual health assistants can help patients find medical information to stay healthy or find medical advice as well as manage appointments or billing fees. Dedicated applications promote healthy behavior and lifestyle in people, while, on the other hand, could help physicians and healthcare professionals to better understand patients’ needs by providing good feedback and to improve guidance and support throughout daily life. In particular, with conditions such as dementia or reduced mobility diseases, robots could help in keeping the body and mind fit.

Biomimetic robotics integrates biomimetic mechanisms, sensors, and actuators to imitate biological structures and adapt to natural and complex environments. 

BI’s role in hospitals or care systems is invaluable: quite all medical specialties are today supported by AI tools or technologies to improve patient care and management, from genetic to surgery, from diagnosis to biological treatment. 

The increase in AI and BI development and in their roles’ interest is proven by the huge number of results retrieved in major web-based search engines, with over one billion and 370 million results and nearly one million per AI and BI search in Google, respectively (Figure 1).

In summary, the role of BI and robotics in healthcare can be recognized in training, research, early detection, diagnosis, decision-making, treatment, keeping well, end-of-life care.

In the present review, we will focus on the role of BI in vascular surgery in general, and particularly in peripheral artery disease (PAD), and on its chance to substitute human experience in evaluating and treating PAD patients.

## 2. Current Applications and Future Development of BI in Vascular Surgery

### 2.1. Healthcare Information

Medical or health records, as well as images, are useful data to establish the diagnosis, the possible therapeutic options, and the eventual prognosis of vascular patients. The volume of information that could be provided to PAD patients about their disease prevention, treatment, evolution through BI is enormous. Powerful information drives patients towards lifestyle, behavioral, and also environmental favorable changes. Awareness of risk factors for PAD and their influence on the evolution of the disease is an efficacious starting point for acting on the long-term prognosis. For example, in diabetic patients, blood glucose level monitoring could be integrated with therapy strategies adopted for glucose reduction and scheduled PAD laboratory tests to obtain a personalized profile to create a dedicated lifestyle program to decrease the disease progression. Those dedicated programs can be reviewed and implemented by physicians by adding newly acquired resources for treatment. Healthcare providers at different levels have the possibility to easily access patients’ data for monitoring and reassessing treatment plans and programs and for communicating with patients. Changes in lifestyle are the natural evolution of such programs, which have been applied with particular efficacy for coronary artery disease patients in developed countries.

### 2.2. Detection and Characterization of Disease

BI can surely be employed in defining disease prediction models by combining different data sources like demographics, medical records, and angiographic features, thus decreasing the possibility of under-recognized detection of vascular disease. Supervised Learning (SL) and Natural Language Processing (NLP) are examples of BI processes to maximize accuracy in vascular disease detection [2]. In their experience on 1569 patients’ data, Afzal and colleagues [2] developed a natural language processing system for automated ascertainment of PAD cases from clinical narrative notes. They made a comparison between the performance of the NLP algorithm with billing code algorithms, using ankle–brachial index test results as gold standards, and found that NLP had greater accuracy in PAD diagnosis [2]. Combining genetic information with other dataset sources to identify and characterize the disease is the new and challenging frontier of BI. It has been applied to AAA identification [3], where personal genomes and electronic health record (EHR) data were integrated to develop a machine learning framework, but can be more challenging in other vascular diseases, such as PAD, because of the difficulty of identifying unique genetic risks. A genome-wide association study of PAD in the Million Veteran Program was performed by testing more than 32 million DNA sequence variants with PAD in veterans and replicating those results in PAD cases and controls in the UK Biobank [4]; among 19 PAD genetic loci, the authors identified four PAD-specific and eleven associated with disease in coronary, cerebral, and peripheral beds, highlighting the difficulty to identify genetic specificity of PAD. Increasing the number of datapoints utilized might possibly overcome this limitation in the future [4]. In their investigative work, Tang et al. [5] were able to predict mortality rate risk in coronary artery bypass grafting (CABG) patients using AI and big data technologies in a risk prediction model. In 2013, Ruel and colleagues [6] had reported on the use of minimally invasive CABG (MICS CABG) and had predicted the widespread robot use in cardiovascular disease treatment. It is to be noted that a comprehensive review published in 2023 [7] has also coined the abbreviation AISER to indicate artificial intelligence, computational simulations, and extended reality that, among all new technologies, are making the greater impact on the healthcare system. The authors highlighted the expected improvement in accuracy, standardization, speed, cost, and accessibility of AI, computational simulations, and extended reality in the near future. Major studies investigating AI and BI in PAD are summarized in Table 1.

### 2.3. Automatic Image Analysis

Automatic analysis of images and videos is currently achieved by Computer Vision in numerous medical branches. In vascular diseases, computational carotid plaque composition and stenosis analysis [8,9,10,11,12], automatic detection and characterization of ischemic brain lesion [13], 3-dimensional analysis of aneurysms morphology or post-endovascular repair endoleak surveillance [14,15,16,17,18], duplex scan (DS) or computed tomography angiography (CTA), and magnetic resonance imaging identification, localization, and quantification of PAD disease [19,20,21] are examples of BI tools for optimizing surgical or endovascular strategies. Similarly, coronary vessels 3-dimensional analysis has enhanced surgical and endovascular treatment options. An increasing number of deep learning models are being developed to help image analysis and interpretation in a wide variety of diseases. A recently published systematic review on BI support for clinical decision in acute ischemic stroke identified one hundred twenty-one publications proposing a clinical decision support system using BI techniques [13]. In PAD, Luo and colleagues [19] and Dai and colleagues [20] developed BI-based models to automate the interpretation of ultrasound and tomographic studies in affected patients, respectively. Both authors reported reliability and accuracy of machine learning and convolutional neural network models in diagnosis of PAD [19,20]. 

### 2.4. Natural Language Processing Model for Retrieve Patients’ Disease

Natural language processing models have been employed to analyze and code medical reports by using linear classifiers and neural networks. They can be used for large-scale retrospective patient identification to develop community, as well as personalized, prevention and treatment strategies. An accuracy of more than 90% in predicting history and presence of disease has been demonstrated in carotid pathology by natural language processing models, gaining efficiency for large-scale patient identification and monitoring [22]. Moreover, they can be used to develop follow-up protocols and future research paths.

### 2.5. Personalized Medical Decision-Making

Machine learning models are of the utmost importance in driving choice between different new and old therapies, often with multiple drugs whose interactions should be adequately investigated to reduce side effects. Analysis of costs could also be machine-generated, thus comparing clinical, as well as economic, risks and benefits, helping in generating personalized medicine and implementing safety surveillance. 

### 2.6. Risk Stratification

BI’s capability of combining hundreds of different risk factors for vascular diseases, with coronary disease and PAD in particular, increases the possibility of identifying complex relationships that can make the diseases more or less aggressive in their clinical features. A typical example is the ML algorithms that can predict mortality risk in PAD patients [23], or rate of major adverse cardiovascular events estimation, based on risk factors and vasculature imaging detection and analysis [24,25,26,27,28,29,30]. The increased capability of PAD classification and prediction of ML approaches respect for stepwise logistic regression models has been reported some years ago [23]. Mortality risk is also assessed and evaluated by those approaches and can be used to develop large-scale prevention programs. Detection of diabetic retinopathy in retinal fundus analysis is another example of deep machine learning use for risk detection and stratification [26,28]. Skandha and Saba [31,32] reported on an AI software developed by Suri et al. for characterization of carotid duplex scan and detection of symptomatic and asymptomatic carotid stenosis for stroke prediction, which could be increasingly trained by new datasets, thus increasing the potential for prevention. 

### 2.7. Surveillance Protocols and Patterns

By combining data on surveillance protocols and events, BI is able to predict the rate of adverse events in vascular patients and thus to suggest different surveillance or treatment strategies at different time points. An example is the use of BI algorithms to predict aneurysmal growth and rupture that could help in refining risk stratification and therefore developing different surveillance patterns and strategies [33,34,35]. By localizing and segmenting the aneurysmal thrombus, a polygonal aneurysmal model is generated and aneurysm geometry is used to set individual patient surveillance protocols in order to monitor future growth. Automatic quantitative measurements and morphologic characterization are nowadays standard AI tools for abdominal aortic aneurysm management [36] (Figure 2). 

### 2.8. Research in Evidence-Based Medicine (EBM)

Combination and analysis of what is called “big data” can tackle EBM matters in few seconds. From the basic biological to the clinical research, AI and BI could promote useful information extraction, analysis, and manipulation. BI patterns have been created to help in clinical trials’ design and data collection, together with their analysis of inclusion and exclusion criteria and results. The combination of those results in metanalysis is an easy and correct way to help in finding the correct EBM path for patients. Moreover, current practice databases could be used by AI to derive rapid decision-making steps for clinical practice, like in Stanford Medicine’s “green button service”, a tool that analyzes Stanford’s medical records data and delivers a consultation report back to physicians, thus allowing them to obtain quick access to evidence on which treatment works for which patients and also by emulating a familiar physician workflow, at the same time keeping experts in the loop [37]. The role of BI and AI in EBM is rapidly increasing over time: it is of note that, nowadays, all scientific journals use a note of caution for all authors not to use BI in writing their manuscripts, since nowadays in writing BI might sometimes outpace human capabilities.

### 2.9. Robots for Care, Surgery, or Drug Administration

Robots are increasingly employed in daily activities in industrial countries. It is not surprising that BI machines are employed to take care of patients, since their role in medicine has increased in the last 30 years, from laboratory testing to highly complex surgical procedures, from rehabilitation or physical therapy performances to delivering drugs in the correct time window. Robot-guided surgery has increased significantly in the last years, so much so that nowadays some interventions can be performed quite entirely by AI-assisted robots. CTA and fluoroscopic images are continuously integrated by AI software to adjust robotic movements to physiologic and pathologic changes in the human body. Newly developed biomimetic robots are being applied in our daily life, starting from the humanoid ATLAS (Boston Dynamics, Figure 3), demonstrating human-level agility but also biologically designed to fulfill different tasks. 

### 2.10. Remote Patients’ Care

Invaluable tools for remote care are AI-derived. Imaging post-processing and automatic enhancement and measurements could help in promoting remote guidance of healthcare professionals. This applies to basic electrocardiographic readings or to more sophisticated imaging processing to deliver remote care (Figure 4). The possibility to apply remote control to medical activities has sensibly increased patients’ care in less-developed areas. This has a deep impact on the development and perception of health systems all around the world. Thanks to new AI- and BI-derived algorithms and to remote control, distant districts’ patients could also receive the best support in time-critical diseases. The use of AI-derived diagnostic tools could enhance a quick diagnosis by major experts in any particular medical field. BI and AI supports are also crucial for remote treatment whenever needed. Finally, and once again, robots are currently being more and more employed for remote intervention, especially for surgery or interventional procedures, as in the daVinci Robotic Surgical System.

### 2.11. Education and Training of Surgeons

Simulation can sometimes be a good surrogate for reality. Recently, sophisticated and realistic virtual reality simulators have been developed and tested for surgery or endovascular training. The capability of combining images and operating technologies has sensibly improved surgical horizons. Their usefulness has been proved in numerous studies, especially in endovascular procedures [38]. Moreover, today’s training can be conducted quite everywhere, with BI helping through smartphones in short catch-up sessions always usable.

## 3. Peripheral Arterial Disease (PAD) and BI

Prevalence of PAD has been estimated to be more than 200 million occurrences in the whole world in 2015 [39], a figure probably underestimated. Chronic limb-threatening ischemia (CLTI) represents PAD’s last stages, and occurrences have undoubtfully increased over the last years, together with the rise in prevalence of diabetes and population ageing, so much so that nowadays wound care accounts for 2–4 per cent of European healthcare costs [40]. Prompt diagnosis of PAD, even in the pre-surgical phase, is of the utmost importance to start preventive treatments that could help save limbs, together with lives. The history of PAD treatment has gained a dramatically important evolution in the endovascular era, to such an extent that nowadays the major classification of the disease is endovascularly driven (GLASS classification) [41]. 

At present, AI and BI may play a central role in developing new diagnostic and treatment solutions in the vascular field, given the possibility of accessing clinical, biological, and imaging data. Vascular disease management by BI could provide the foundation for diagnostic, prognostic, and operative solutions. The possibility of assessing the vascular anatomy in every patient, as well as the burden of atherosclerosis, and thus classifying the level and degree of the disease, as it is currently performed in aortic aneurysm characterization, is surely charming. This possibility, combined with that of correctly defining the perioperative complications risk, could guide surgeons in the choice of the best surgical technique. Sizing and planning of the best endovascular treatment is enhanced by BI, which could lead to define a personalized therapeutic strategy for every patient. CTA automatic peripheral vessel identification to localize and quantify disease has been reported in a study by Dai et al. [20], where they used a convolutional neural network based on 17,050 axial images in 265 PAD patients to classify above- and below-knee artery stenosis with an accuracy greater than 90% in the majority of stenosis classes. Also, Doppler waveforms can be integrated in neural networks to detect and classify PAD with a reported accuracy of 0.69, 1, and 0.86 for mono-, bi-, and triphasic waveforms, respectively [19]. 

It is not to be overlooked that PAD treatment is often performed between different specialties, with the difficulty of integrating experiences and resources. AI and BI can play a significant role in integrating objective evaluation tools and offering evidence-based solutions. Automating screening of patients’ medical records can help in identifying latent PAD, thus driving towards more specific diagnostic tools to be used in order to classify the burden of the disease and help in making the best treatment choice by integrating clinical events’ information [23,42,43]. Natural Language Processing algorithms are the best representation of BI-generated classification of PAD [44]. Sometimes, given the broad heterogeneity of presentation of PAD despite the commonly known risk factors, Unsupervised Learning, mainly based on clustering algorithms of multidimensional patient data, may help in revealing disease subgroups with different phenotypes and in tailoring treatment strategies in those different subgroups. Machine learning, by incorporating information from different areas of human life, such as behavior and lifestyle, could be employed to generate predictions and recommendations for optimal life changes (e.g., regular physical activity) that can influence the development and progression of arterial disease [45]. It could also be used to promote and maintain a supervised exercise therapy (SET) program in PAD patients, as established by the Society for Vascular Surgery in 2021 [46]. Differential diagnosis could also be improved by achieving the so-called precision cardiovascular medicine, whose ultimate aim is, of course, to choose the right treatment for the right patient at the right time. Mortality risk prediction, as well as amputation-free survival estimation and surgical site infection risk after revascularization, can be calculated using ML algorithms that combine data on a huge number of different risk factors and can predict their effects and future developments [23,47]. Unfortunately, ML-based major adverse limb events prognostic tools are currently not being developed, mainly because necessary datasets for the algorithms’ creation are difficult to obtain. Reliability of certain kinds of databases might be debatable when a prediction tool must be developed; likewise, physiological markers detected during physical examination are difficult to classify, categorize, and report. Hence, these data cannot be reliably incorporated and used for ML constructions. Nevertheless, as anticipated from coronary CTA [48,49], ML models might be trained by specifically built databases to establish the relationship between vessel CTA images and symptoms, or angiosome lesions. The key point is, once again, which dataset to provide the ML algorithm with. This dataset must be tailored to specifically satisfy the required need. This undoubtedly poses difficulties in compiling and maintaining the dataset. Despite some studies demonstrating the capability of detecting the severity of PAD, differentiating ischemic vs. neurogenic claudication, predicting PAD mortality, ambulation potential after amputation, and surgical site infection following lower extremity bypass, by integrating imaging and clinical functional data [50], PAD presents itself with a broad heterogeneity in clinical features and outcomes, so that traditional physical examination in experienced hands is crucial to the right choice between whether, how, and when to treat. BI has nowadays the potential for improving and eventually replacing the traditional tools for cardiovascular diseases management in the biomimetic nanoparticles technology. 

## 4. Limitations and Risk of Bias

Apart from the risk of mythizing AI and BI, some limitations to its applicability and role must be acknowledged. Big amounts of data are mandatory in order to create algorithms and neural networks. These data must be correctly collected with a potential risk of misclassification and misinterpretation. Data heterogeneity can be responsible for inconclusive or also paradoxical AI- and BI-driven solutions. Clinical registries and trials are of the utmost importance to create a reliable database for BI algorithms’ establishment.

The algorithms used to generate diagnostic and therapeutic tools may lack the possibility to establish causal relationship. In addition, interpretability and accountability of such algorithms can also be debatable in certain cases. Physicians’ expertise in BI algorithms’ interpretation is hardly replaceable [51]. Implementation of physicians’ and data scientists’ collaboration will be required in the near future, to learn to ‘speak the same language’. Lastly, new AI tools require adequate computational platforms and power that might not be available in every environment.

## 5. Conclusions

AI and BI in particular are gaining increasing importance in patients’ diagnostics, treatment, surveillance, and research. They also have a high potential for future development in vascular medicine and PAD in particular, where integration of clinical and imaging modalities, together with a clinician’s experience, are still the best combination for PAD diagnosis and treatment nowadays.

To develop BI diagnostic and caring tools, and transform them into real-world practice, will be the focus of future technology implementation and use in daily routine. 

## Figures and Tables

**Figure 1 biomimetics-09-00465-f001:**
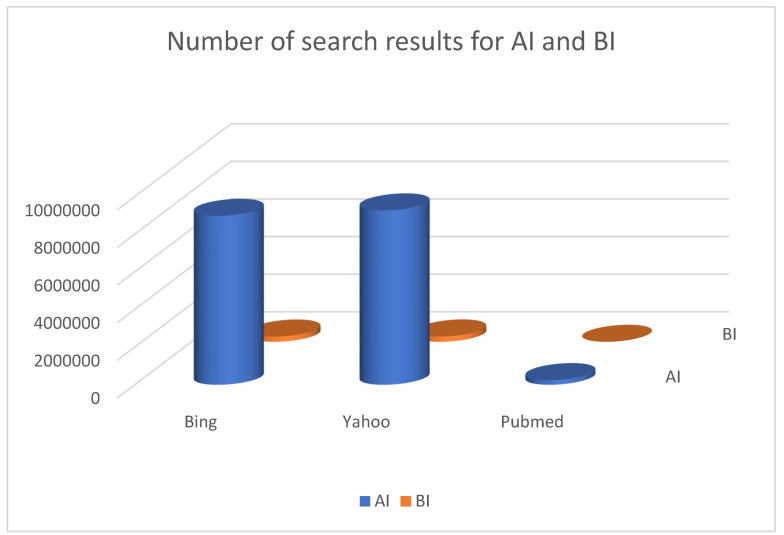
Search results for terms “Artificial Intelligence-AI” and “Biomimetic Intelligence-BI” on major web and medical search engines in June 2024.

**Figure 2 biomimetics-09-00465-f002:**
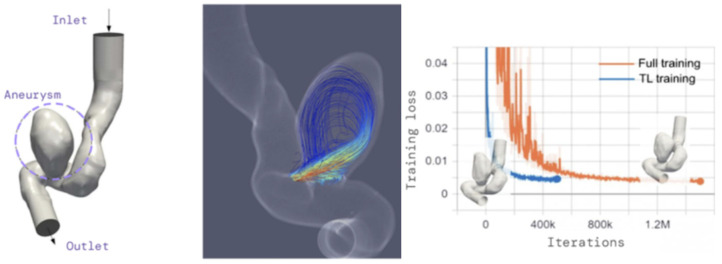
SimNet flow-field simulation according to geometry of an intracranial aneurysm. Reproduced with permission by Olexandre Isayev.

**Figure 3 biomimetics-09-00465-f003:**
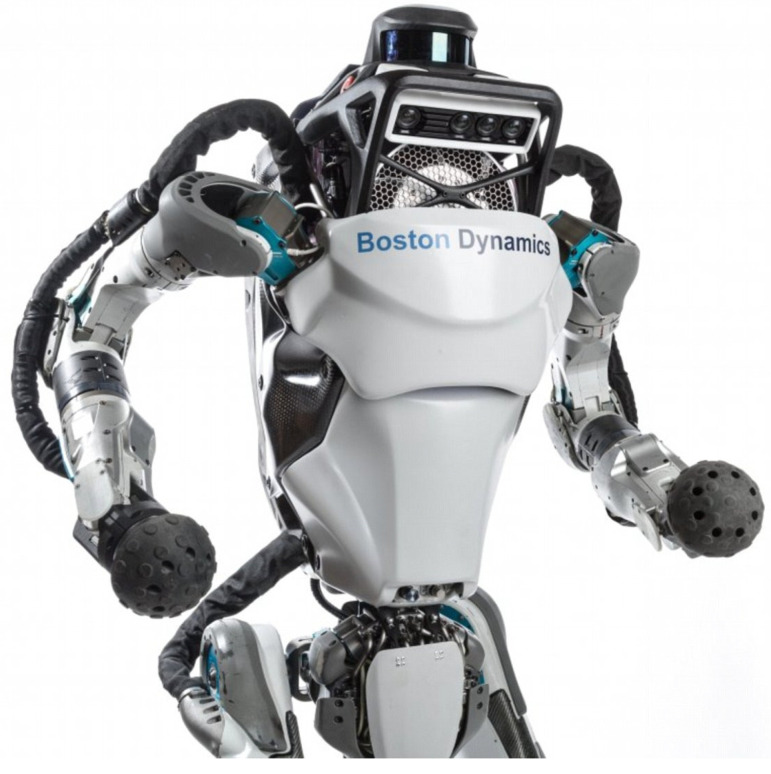
ATLAS, the humanoid robot developed by BostonDynamics, Inc. (www.bostondynamics.com, Waltham, Massachusetts, USA; URL accessed on 3 July 2024) has advanced human capabilities and lifelike movements.

**Figure 4 biomimetics-09-00465-f004:**
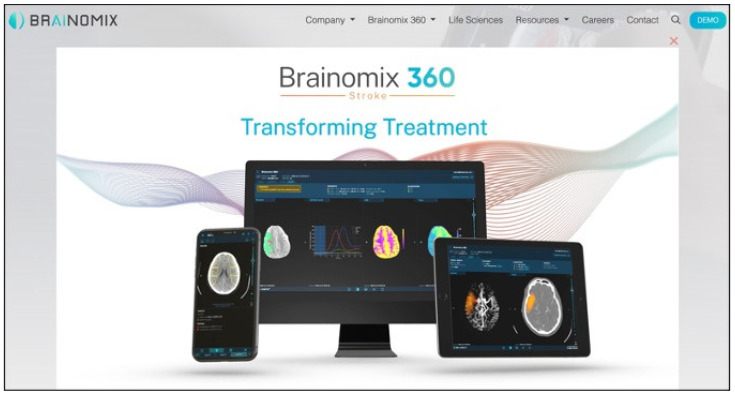
BrAInomix^®^ is a dedicated AI-based software that is able to rapidly analyze brain CT to calculate ASPECTS score, perfusion brain CT to calculate penumbra and core ischemia, and angio-CT to calculate ischemic areas and collateral pathways.

**Table 1 biomimetics-09-00465-t001:** Summary of studies investigating AI and BI application in peripheral arterial disease (PAD), coronary artery disease (CAD), cerebral vascular disease, and abdominal aortic aneurysm (AAA) disease.

Author	Disease Investigated	Source Used	Results
Afzal et al. [2]	PAD	Clinical narrative notes	NLP system greater accuracy in PAD diagnosis
Li et al. [3]	AAA	Electronic health record and personal genomes	Identification of disease
Klarin et al. [4]	PAD, CAD, cerebral vascular disease	Million Veteran Program DNA sequences and UK Biobank	Identification of disease-specific genetic loci
Tang et al. [5]	CAD	Clinical CABG patients’ data	Surgical risk prediction model construction

## Data Availability

The original contributions presented in the study are included in the article, further inquiries can be directed to the corresponding authors.

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
