# Peer review of "Current Applications and Future Perspectives of Artificial and Biomimetic Intelligence in Vascular Surgery and Peripheral Artery Disease"

_biomimetics, 2024, doi:10.3390/biomimetics9080465_

Round 1
Reviewer 1 Report
Comments and Suggestions for Authors
The study titled "CURRENT APPLICATIONS AND FUTURE PERSPECTIVES OF ARTIFICIAL INTELLIGENCE IN VASCULAR SURGERY AND PERIPHERAL ARTERY DISEASE" is well-written. They mentioned the application of AI in PAD and vascular surgery. I have some comments for improvement:
1- Define abbreviations in their first use and make sure that abbreviated forms are being used after the definition.
2- Add data about the use of machine learning in other cardiovascular interventions like PCI and CABG as additional data for drawing more significant conclusions.
3- I found some typos and grammatical errors.
4- Add a figure and/or table summarizing the existing literature on the topics you have investigated in your review.
5- Add the clinical utility of your findings for a cardiologist/cardiac surgeon.
Author Response
The authors want to thank the Editor and the Reviewers for their invaluable work that help us to improve our manuscript. Following are the detailed responses to your comments and suggestions.
Reviewer 1
The study titled "CURRENT APPLICATIONS AND FUTURE PERSPECTIVES OF ARTIFICIAL INTELLIGENCE IN VASCULAR SURGERY AND PERIPHERAL ARTERY DISEASE" is well-written. They mentioned the application of AI in PAD and vascular surgery. I have some comments for improvement:
- Define abbreviations in their first use and make sure that abbreviated forms are being used after the definition.
Thank you for pointing this out. The manuscript has been edited accordingly.
- Add data about the use of machine learning in other cardiovascular interventions like PCI and CABG as additional data for drawing more significant conclusions.
Thank you for your suggestion. The text has been enriched with new cardiological-dedicated citations (numbers 5 to 7).
- I found some typos and grammatical errors.
The text has been revised.
- Add a figure and/or table summarizing the existing literature on the topics you have investigated in your review.
A Figure reporting searching results in web search engines has been added in the text.
- Add the clinical utility of your findings for a cardiologist/cardiac surgeon.
Thanks for your suggestion. The text has been edited accordingly.
Reviewer 2 Report
Comments and Suggestions for Authors
The manuscript presents a comprehensive overview of the integration of Artificial Intelligence (AI) in healthcare, particularly in the management of Chronic Limb-Threatening Ischemia (CLTI), a severe stage of Peripheral Artery Disease (PAD). It outlines the evolution of AI since its inception in 1956 and categorizes its functions into Machine Learning (ML), Deep Learning (DL), Artificial Neural Network (ANN), Convolutional Neural Network (CNN), and Computer Vision (CV).
The review highlights the increasing incidence of CLTI, correlating with the rise in diabetes and an aging population. It introduces Biomimetic Intelligence (BI), which utilizes natural system principles to develop biological algorithms for various applications. The main topic of the MS focuses on AI and BI's potential to revolutionize diagnostics and treatment in the vascular field by leveraging clinical, biological, and imaging data. It emphasizes AI's capability to analyze vascular anatomy, assess atherosclerosis burden, classify disease severity, and plan optimal endovascular treatments and surgical procedures.
I commend the paper for addressing the significant impact of AI and BI on the vascular field. It challenges traditional reliance on physician experience with data-driven insights that could enhance patient outcomes in PAD treatment. The MS suggests a promising direction for future healthcare, where technology and human expertise converge to improve the quality of care. However, major revisions are needed to improve this review because most sections only cite limited references. :
- In general, section 2 is underdeveloped. For a review, the authors should include more discussions, cite more studies, etc.
- Section 2.2 will benefit from a table outlining the model specs.
- Section 2.10 demands more insights from the literature/applications.
- Proofreading is also needed in the MS.
Proofreading is needed- minor review
Author Response
The authors want to thank the Editor and the Reviewers for their invaluable work that help us to improve our manuscript. Following are the detailed responses to your comments and suggestions.
Reviewer 2
The manuscript presents a comprehensive overview of the integration of Artificial Intelligence (AI) in healthcare, particularly in the management of Chronic Limb-Threatening Ischemia (CLTI), a severe stage of Peripheral Artery Disease (PAD). It outlines the evolution of AI since its inception in 1956 and categorizes its functions into Machine Learning (ML), Deep Learning (DL), Artificial Neural Network (ANN), Convolutional Neural Network (CNN), and Computer Vision (CV).
The review highlights the increasing incidence of CLTI, correlating with the rise in diabetes and an aging population. It introduces Biomimetic Intelligence (BI), which utilizes natural system principles to develop biological algorithms for various applications. The main topic of the MS focuses on AI and BI's potential to revolutionize diagnostics and treatment in the vascular field by leveraging clinical, biological, and imaging data. It emphasizes AI's capability to analyze vascular anatomy, assess atherosclerosis burden, classify disease severity, and plan optimal endovascular treatments and surgical procedures.
I commend the paper for addressing the significant impact of AI and BI on the vascular field. It challenges traditional reliance on physician experience with data-driven insights that could enhance patient outcomes in PAD treatment. The MS suggests a promising direction for future healthcare, where technology and human expertise converge to improve the quality of care. However, major revisions are needed to improve this review because most sections only cite limited references. :
- In general, section 2 is underdeveloped. For a review, the authors should include more discussions, cite more studies, etc.
Thank you for your suggestion. Some sections have been implemented.
- Section 2.2 will benefit from a table outlining the model specs.
Thank you for your suggestion. Table I has been added in the text.
- Section 2.10 demands more insights from the literature/applications.
Thank you for your suggestion. This section has been implemented.
Proofreading is also needed in the MS.
The text has been revised.
Round 2
Reviewer 1 Report
Comments and Suggestions for Authors
I have no additional comments.